# The acid ceramidase/ceramide axis controls parasitemia in *Plasmodium yoelii*-infected mice by regulating erythropoiesis

Anne Günther[1], Matthias Hose[1], Hanna Abberger[1], Fabian Schumacher[2], Ylva Veith[3], Burkhard Kleuser[2], Kai Matuschewski[3], Karl Sebastian Lang[4], Erich Gulbins[5], Jan Buer[1], Astrid M Westendorf[1], Wiebke Hansen[1]*

[1]Institute of Medical Microbiology, University Hospital Essen, University Duisburg-Essen, Essen, Germany; [2]Institute of Pharmacy, Freie Universität Berlin, Berlin, Germany; [3]Molecular Parasitology, Institute of Biology/Faculty for Life Sciences, Humboldt University Berlin, Berlin, Germany; [4]Institute of Immunology, University Hospital Essen, University Duisburg-Essen, Essen, Germany; [5]Institute of Molecular Biology, University of Duisburg-Essen, Essen, Germany

**Abstract** Acid ceramidase (Ac) is part of the sphingolipid metabolism and responsible for the degradation of ceramide. As bioactive molecule, ceramide is involved in the regulation of many cellular processes. However, the impact of cell-intrinsic Ac activity and ceramide on the course of *Plasmodium* infection remains elusive. Here, we use Ac-deficient mice with ubiquitously increased ceramide levels to elucidate the role of endogenous Ac activity in a murine malaria model. Interestingly, ablation of Ac leads to alleviated parasitemia associated with decreased T cell responses in the early phase of *Plasmodium yoelii* infection. Mechanistically, we identified dysregulated erythropoiesis with reduced numbers of reticulocytes, the preferred host cells of *P. yoelii*, in Ac-deficient mice. Furthermore, we demonstrate that administration of the Ac inhibitor carmofur to wildtype mice has similar effects on *P. yoelii* infection and erythropoiesis. Notably, therapeutic carmofur treatment after manifestation of *P. yoelii* infection is efficient in reducing parasitemia. Hence, our results provide evidence for the involvement of Ac and ceramide in controlling *P. yoelii* infection by regulating red blood cell development.

*For correspondence:
wiebke.hansen@uk-essen.de

Competing interest: The authors declare that no competing interests exist.

## Editor's evaluation

This study provides evidence that murine acid ceramidase (Ac) is required for normal erythropoiesis and the development of rodent malaria. The findings are of interest in understanding molecular processes involved in regulating erythropoiesis, as well as the potential to develop host-directed therapies for malarial parasites that target human reticulocytes.

## Introduction

Malaria remains one of the most life-threatening infectious diseases in the world, with estimated 241 million cases and 627,000 related deaths in 2020 (***World health organization, 2021***). Malaria is caused by infections with the parasite *Plasmodium* that undergoes a complex life cycle within its hosts. In the mammalian host, plasmodia invade red blood cells (RBCs) for their asexual propagation. Depending on the species, the parasites favor different RBC maturation stages for their

intraerythrocytic development. While the most lethal human malaria species, *Plasmodium falciparum* is able to infect both, mature erythrocytes and immature reticulocytes, the most widespread parasite *Plasmodium vivax* is restricted to invade the latter (*Garnham, 1966*; *Kitchen, 1938*; *Chwatt, 1948*). The formation of RBCs, known as erythropoiesis, primarily takes place in the bone marrow, where hematopoietic stem cells give rise to multipotent progenitors that stepwise differentiate into erythroblasts and reticulocytes (*Moras et al., 2017*). Early reticulocytes egress from the bone marrow and constitute 1–3% of circulating RBCs before they develop into mature erythrocytes (*Chin-Yee et al., 1991*). During erythropoiesis, the cells start to produce hemoglobin, lose nuclei and organelles, and induce plasma membrane remodeling. While the erythroid lineage marker Ter119 is induced in erythroblasts and subsequently persistently expressed, expression of the transferrin receptor 1 (CD71) gets lost during the transition from reticulocytes to mature erythrocytes (*Chen et al., 2009*).

Several metabolic processes are involved in the regulation of erythropoiesis and therefore might have an impact on *Plasmodium* infection. The best-known example of metabolites essential for erythropoiesis is circulating iron, which is almost exclusively absorbed by erythropoietic demand. Plasma concentration, tissue distribution, and absorption are under the control of hepcidin (*Kautz and Nemeth, 2014*). Upon *Plasmodium* blood stage infection, hepcidin expression in the liver is upregulated, which in turn modulates parasitemia, disease outcome, and superinfection (*Howard et al., 2007*; *Wang et al., 2011*; *Portugal et al., 2011*; *Spottiswoode et al., 2014*), fully supporting clinical studies that found associations of asymptomatic *P. falciparum* and *P. vivax* parasitemia with increased hepcidin concentrations and anemia (*de Mast et al., 2010*; *Casals-Pascual et al., 2012*). Additional essential metabolic processes include fatty acid oxidation to preserve renewal of hematopoietic stem cells, amino acid-induced mTOR signaling, and glutamine metabolism (*Oburoglu et al., 2016*). For terminal erythropoiesis, alterations in the lipid metabolism with reduced phosphatidylcholine and enhanced sphingomyelin have been proposed to play a critical role (*Huang et al., 2018*). However, the effects of these metabolic processes on *Plasmodium* blood infection remain elusive.

During the erythrocytic stage of malaria, CD4[+] T cells play a crucial role in regulating the immune response. By secreting IFN-γ, they activate other immune cells, for example, macrophages, which are necessary for parasite clearance and control. Moreover, CD4[+] T cells provide help for B cells to induce protective responses against the blood-stage parasites (*Perez-Mazliah and Langhorne, 2014*). Recently, we showed that modulations of the sphingolipid (SL) metabolism enhance T cell activity during *Plasmodium yoelii* infection (*Hose et al., 2019*). SLs are important components of biological membranes. Besides their importance in cell integrity, they have drawn attention as bioactive molecules, playing a crucial role in cell function and fate (*Hannun and Obeid, 2018*). The SL metabolism consists of a network of various enzymes that catalyze the formation and degradation of the different SLs. Ceramide represents a central hub in this SL network. It can either be formed by de novo synthesis or by metabolism of complex SLs, for example, sphingomyelin (*Hannun and Obeid, 2008*). Ceramidases hydrolyze ceramide to sphingosine, which can be further phosphorylated by sphingosine kinases, forming sphingosine-1-phosphate (S1P). So far, five different human ceramidases have been described according to their pH optimum: acid ceramidase (Ac, N-acylsphingosine amidohydrolase 1, *ASAH1*), neutral ceramidase (Nc, *ASAH2*), and alkaline ceramidase 1–3 (Acer1-3, *ACER1-3*), each having a murine counterpart (*Mao and Obeid, 2008*). Ac is localized in the lysosomes, where it preferentially hydrolyzes ceramides with medium-chain length (C12–C16) (*Ferlinz et al., 2001*; *Momoi et al., 1982*). Mutations in the *ASAH1* gene can lead to two different rare diseases, the lysosomal storage disorder Farber disease (FD) and an epileptic disorder called spinal muscular atrophy with progressive myoclonic epilepsy (SMA-PME) (*Yu et al., 2018*). Ac and ceramide have also been implicated to play a role in several infectious diseases. While the pharmacological inhibition of Ac by ceranib-2 decreases measles virus replication in vitro (*Grafen et al., 2019*), genetic overexpression of Ac rescues cystic fibrosis mice from pulmonary *Pseudomonas aeruginosa* infections in vivo (*Becker et al., 2021*). *Lang et al., 2020* recently showed that Ac expression in macrophages limits the propagation of herpes simplex virus type 1 and protects mice from severe course of infection. However, the role of host cell-intrinsic Ac activity and ceramide during *Plasmodium* infection remains elusive.

Here, using the nonlethal rodent *Plasmodium* strain *P. yoelii* 17XNL as murine malaria model, we evaluated how Ac ablation and pharmacological inhibition modulate the course of infection in mice. We demonstrate that increased ceramide levels in Ac-deficient mice or treatment with the Ac inhibitor

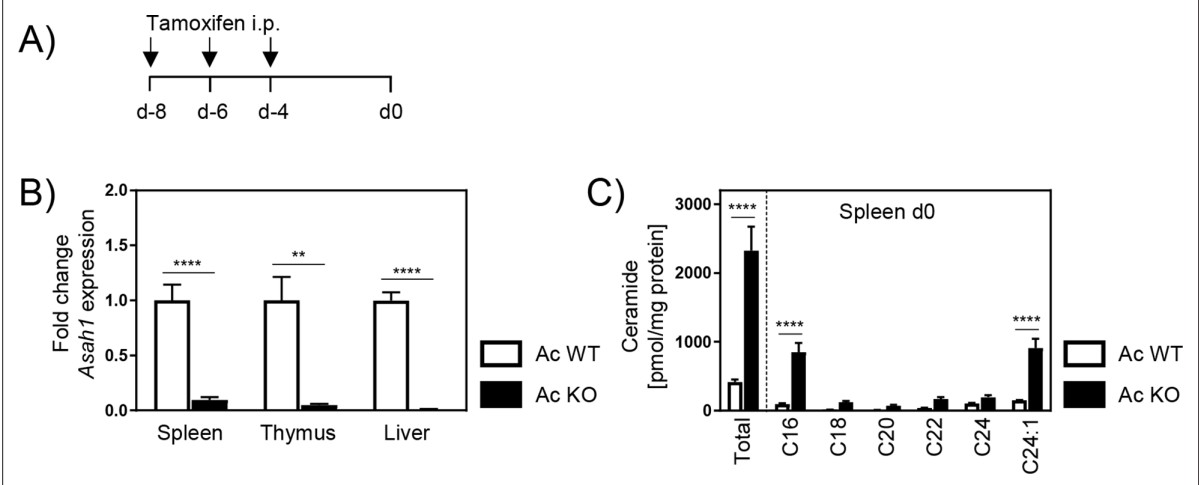

**Figure 1.** Ceramide accumulation in acid ceramidase (Ac) knockout (KO) mice. (**A**) Induction of Ac deficiency in *Asah1/Rosa26^creER/+* mice (Ac KO) was achieved by intraperitoneal injection (i.p.) of 4 mg tamoxifen on days −8, −6, and −4. Tamoxifen-treated *Asah1/Rosa26^+/+* littermates (Ac WT) were used as control. (**B**) Ac KO validation was performed by analyzing *Asah1* mRNA expression via RT-qPCR in spleen, thymus, and liver of Ac KO mice and Ac WT littermates as control on day 0 (n = 5–6). (**C**) Ceramide levels in spleen were determined by high-performance liquid chromatography mass spectrometry (HPLC-MS/MS) (n = 3–4). Data are presented as mean ± SEM. Statistical analyses were performed using Mann–Whitney *U*-test for (**B**) and two-way ANOVA, followed by Sidak's post-test for (**C**) (**p<0.01, ****p<0.0001).

The online version of this article includes the following source data for figure 1:

**Source data 1.** Ceramide accumulation in acid ceramidase (Ac) knockout (KO) mice.

carmofur significantly reduce parasitemia in *P. yoelii*-infected mice. Mechanistically, we identified reduced generation of reticulocytes, the target cells of *P. yoelii*, upon deletion or inhibition of Ac.

## Results

### Ac-deficient mice accumulate ceramide

Since the genetic ablation of Ac is embryonically lethal in mice, we made use of inducible *Asah1/Rosa26^creER/+* (Ac KO) mice, which received tamoxifen 8, 6, and 4 days prior to analysis or *P. yoelii* infection (*Figure 1A*; *Eliyahu et al., 2007*; *Li et al., 2002*). Tamoxifen-treated *Asah1/Rosa26^+/+* (Ac WT) littermates served as control. As depicted in *Figure 1B*, Ac mRNA expression was ubiquitously abolished in all analyzed tissues in Ac KO mice. Consequently, ceramide concentrations determined by mass spectrometry were significantly increased in spleens from Ac KO mice compared to Ac WT mice (*Figure 1C*).

### Ac deficiency results in reduced parasitemia and decreased T cell responses during early stage of *P. yoelii* infection

To study the role of Ac during malaria, we infected Ac KO mice and Ac WT littermates with *P. yoelii*. Interestingly, Ac KO mice showed significantly reduced parasitemia on days 3 and 7 post infection (p.i.) (*Figure 2A*). However, these differences were no longer detectable 10 and 14 days p.i. As splenomegaly is a common clinical signature of *Plasmodium* infections, we determined the spleen weight of infected Ac WT and Ac KO mice. Well in line with the observed effect on parasitemia, spleen weights of Ac KO mice were tremendously decreased compared to those of Ac WT mice 7 days p.i., but no longer differed on day 14 p.i. (*Figure 2B*).

Next, we analyzed the phenotype of T cells from spleen of infected mice 7 days p.i. by flow cytometry. T cells, especially CD4+ T cells, have been described to play an important role in limiting blood-stage malaria (*Perez-Mazliah and Langhorne, 2014*). While frequencies of CD4+ and CD8+ T cells did not differ between *P. yoelii*-infected Ac WT and Ac KO mice, we found significantly increased frequencies of CD4+Foxp3-expressing regulatory T cells (Tregs) in spleen of Ac KO mice compared to Ac WT littermates (*Figure 2C*). Interestingly, CD4+ and CD8+ T cells from Ac KO mice showed decreased expression of marker molecules associated with T cell activation, namely, Ki67, CD49d,

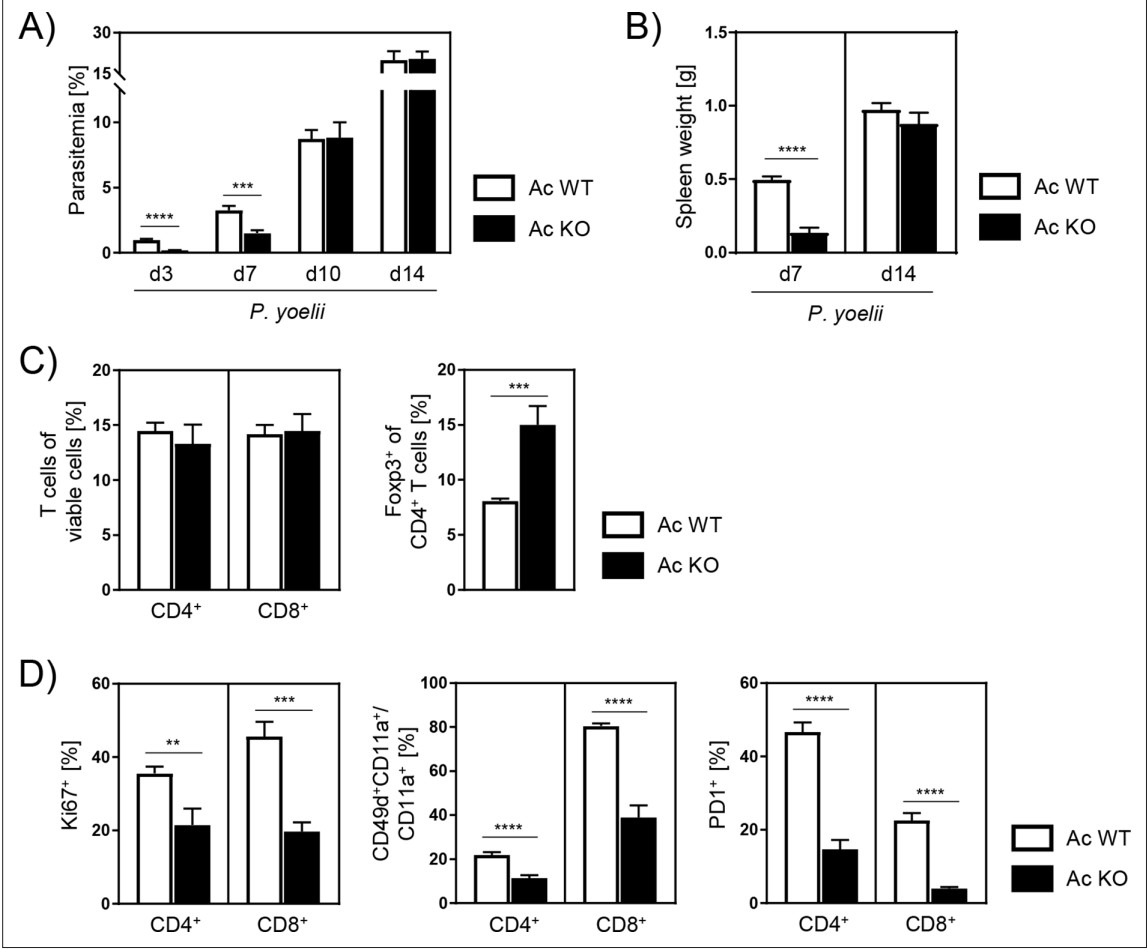

**Figure 2.** *P.yoelii*-infected acid ceramidase (Ac) knockout (KO) mice show decreased parasitemia with less T cell activation in the early phase of infection. (**A**) Parasitemia of *P. yoelii*-infected Ac WT and Ac KO mice was determined at indicated time points by microscopy of Giemsa-stained blood films (n = 9–18). (**B**) Spleen weight on days 7 and 14 post infection (p.i.) (n = 7–9). (**C**) Frequencies of viable CD4+, CD8+, and regulatory T cells (Foxp3+ of CD4+) were analyzed by flow cytometry 7 days p.i. (n = 7–9). (**D**) Percentages of Ki67-, CD49dCD11a/CD11a-, and PD1-expressing CD4+ and CD8+ T cells were determined by flow cytometry 7 days p.i. (n = 7–9). Data from three to five independent experiments are presented as mean ± SEM. Statistical analyses were performed using Mann–Whitney *U*-test for (**A**) and unpaired Student's *t*-test for (**B–D**) (**p<0.01, ***p<0.001, ****p<0.0001).

The online version of this article includes the following source data and figure supplement(s) for figure 2:

**Source data 1.** *P. yoelii*-infected acid ceramidase (Ac) knockout (KO) mice show decreased parasitemia with less T cell activation in the early phase of infection.

**Figure supplement 1.** T cell response of *P. yoelii*-infected acid ceramidase (Ac) knockout (KO) and Ac wildtype (WT) mice 14 days post infection (p.i.).

**Figure supplement 1—source data 1.** T cell response of *P. yoelii*-infected acid ceramidase (Ac) knockout (KO) and Ac wildtype (WT) mice 14 days post infection (p.i.).

CD11a, and PD1 (*Figure 2D*). In accordance with the observations regarding parasitemia and spleen weight, differences were diminished 14 days p.i. (*Figure 2—figure supplement 1*). These results show that Ac deficiency leads to reduced parasitemia in the early phase of *P. yoelii* infection accompanied by diminished T cell responses.

## Cell-specific Ac deletion in T cells and myeloid cells does not affect the course of *P. yoelii* infection

In order to further evaluate the effect of Ac on immune responses during *P. yoelii* infection, we made use of cell type-specific Ac-deficient mice. Since T cells from infected global Ac KO mice were less activated, we next analyzed *Asah1/Cd4^cre* mice that specifically lack Ac expression in CD4+ and CD8+ T cells (*Figure 3A*). However, T cell-specific Ac ablation had no impact on the course of *P. yoelii*

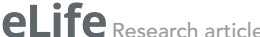

**Figure 3.** T cell-specific and myeloid-specific acid ceramidase (Ac) deletion has no impact on the course of *P. yoelii* infection. (**A**) The knockout of Ac in T cells was confirmed by analyzing *Asah1* mRNA expression of sorted splenic CD4+ and CD8+ T cells from naïve *Asah1/Cd4cre/+* (Ac CD4cre knockout [KO]) mice and *Asah1/Cd4+/+* littermates (Ac CD4cre wildtype [WT]) as controls via RT-qPCR (n = 2–4). (**B**) Parasitemia (left panel) and spleen weight (right panel) of *P. yoelii*-infected Ac CD4cre KO mice and Ac CD4cre WT littermates was determined at indicated time points (n = 7–10). (**C**) The knockout of Ac in myeloid cells was confirmed by analyzing *Asah1* mRNA expression of macrophages, dendritic cells, and neutrophils isolated from spleen, peritoneal lavage (pLavage), and blood of naïve *Asah1/Lyz2cre/+* (Ac Lyz2cre KO) mice and *Asah1/Lyz2+/+* littermates (Ac Lyz2cre WT) as controls via RT-qPCR (n = 2–6). (**D**) Parasitemia (left panel) and spleen weight (right panel) of *P. yoelii*-infected Ac Lyz2cre KO and Ac Lyz2cre WT mice was determined at indicated time points (n = 9). Data from two independent experiments each are presented as mean ± SEM.

The online version of this article includes the following source data for figure 3:

**Source data 1.** T cell-specific and myeloid-specific acid ceramidase (Ac) deletion has no impact on the course of *P. yoelii* infection.

infection as Ac CD4cre KO (*Asah1/Cd4cre/+*) and Ac CD4cre WT (*Asah1/Cd4+/+*) mice showed similar spleen weights and parasitemia 7 and 14 days p.i. (*Figure 3B*). We were wondering whether an early enhanced innate immune response as first line of defense contribute to an improved parasite clearance in global Ac KO mice and therefore to less T cell activation in the initial phase of infection. Thus,

we used *Asah1/Lyz2^cre* mice, which exhibit decreased Ac expression in myeloid cells, for example, macrophages, dendritic cells, and neutrophils (**Figure 3C**). However, upon *P. yoelii* infection, Ac Lyz2cre KO (*Asah1/Lyz2^cre/+*) mice and Ac Lyz2cre WT (*Asah1/Lyz2^+/+*) littermates did not significantly differ regarding parasitemia and spleen weight (**Figure 3D**).

## Ac KO mice show an impaired erythropoiesis

Since global Ac KO mice showed decreased parasitemia, although cell-specific Ac activity did not alter innate and T cell immune responses during *P. yoelii* infection per se, we analyzed whether Ac

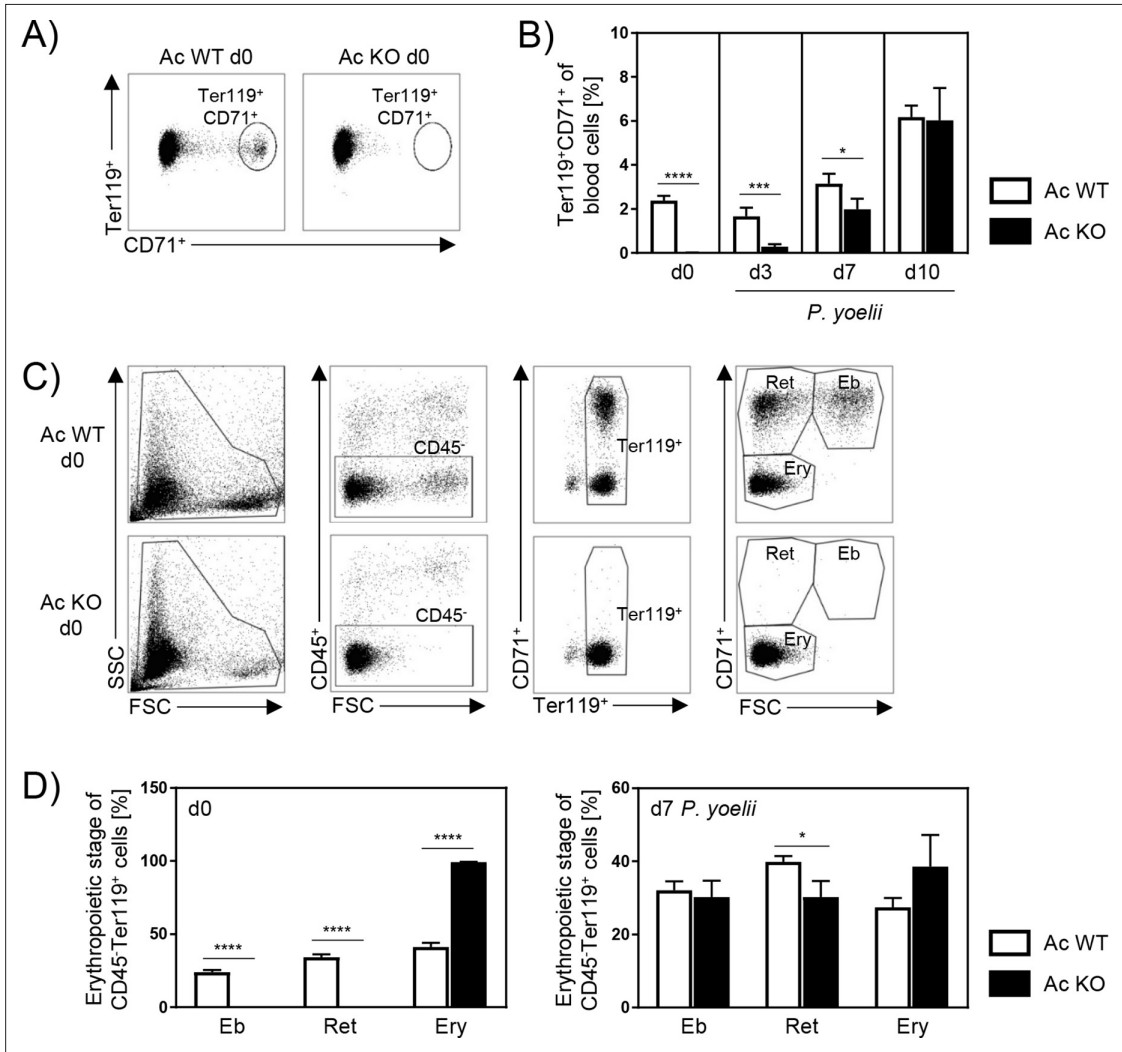

**Figure 4.** Acid ceramidase (Ac) knockout (KO) affects the erythropoiesis. (**A**) Representative flow cytometry dot plot of Ter119+CD71+ reticulocytes in blood of Ac wildtype (WT) and Ac KO mice on day 0. (**B**) Frequencies of reticulocytes in blood of Ac WT and Ac KO mice were determined by flow cytometry on day 0 and after *P. yoelii* infection on days 3, 7, and 10 (n = 8–15). (**C**) Representative flow cytometry gating strategy of erythropoietic developmental stages in bone marrow of Ac WT and Ac KO mice on day 0. Erythropoietic cells were defined as CD45⁻ and Ter119⁺. Depending on size (forward scatter [FSC]) and CD71 expression, cells were defined as erythroblasts (Eb, CD71+FSChigh), reticulocytes (Ret, CD71+FSClow), and mature erythrocytes (Ery, CD71-FSClow). (**D**) Frequencies of different erythropoietic stages in bone marrow of noninfected Ac WT and Ac KO mice on day 0 (left panel) and after *P. yoelii* infection on day 7 (right panel) were analyzed by flow cytometry (n = 6–11). Results from 2–4 independent experiments are presented as mean ± SEM. Statistical analyses were performed using Mann–Whitney *U*-test for (**B**) and unpaired Student's *t*-test for (**D**) (*p<0.05, ***p<0.001, ****p<0.0001).

The online version of this article includes the following source data and figure supplement(s) for figure 4:

**Source data 1.** Acid ceramidase (Ac) knockout (KO) affects the erythropoiesis.

**Figure supplement 1.** Erythropoiesis of noninfected acid ceramidase (Ac) knockout (KO) and Ac wildtype (WT) mice.

**Figure supplement 1—source data 1.** Erythropoiesis of noninfected acid ceramidase (Ac) knockout (KO) and Ac wildtype (WT) mice.

deletion instead has an effect on the host cells of the parasites. Depending on the species, *Plasmodium* parasites prefer immature or mature RBCs for asexual propagation (*Leong et al., 2021*). *P. yoelii* parasites primarily infect reticulocytes, which are defined as Ter119$^+$CD71$^+$ (*Martín-Jaular et al., 2013*). Therefore, we analyzed frequencies of reticulocytes in blood of Ac WT and Ac KO mice by flow cytometry. Strikingly, reticulocytes were dramatically reduced in blood of noninfected Ac KO mice (*Figure 4A*, d0). While they were almost absent on days 0 and 3 p.i., frequencies rapidly increased in Ac KO mice 7 days p.i., resulting in equal percentages of reticulocytes in Ac WT and Ac KO mice 10 days p.i. (*Figure 4B*).

To gain further insights into the impact of Ac expression on erythropoietic cells, we analyzed the development of RBCs within the bone marrow of Ac WT and Ac KO mice in more detail. Depending on size and CD71 expression, erythropoietic cells (CD45$^-$Ter119$^+$) were identified as erythroblasts, reticulocytes, and mature erythrocytes (*Figure 4C*). As depicted in *Figure 4D*, erythroblasts and reticulocytes were hardly detectable in bone marrow of noninfected Ac KO mice on day 0 (left panel). Consistent with raising reticulocyte frequencies in the blood (*Figure 4B*), percentages of erythroblasts and reticulocytes were increasing 7 days p.i. in bone marrow of *P. yoelii*-infected Ac KO mice, but they were still reduced when compared to infected Ac WT mice (*Figure 4D*, right panel). Similar results were obtained in noninfected Ac KO mice (*Figure 4—figure supplement 1*), providing evidence that this phenotype is independent of the infection. Together, these results indicate an impaired erythropoiesis in Ac KO mice, leading to reduced frequencies of reticulocytes the main target cells of *P. yoelii* parasites.

## Ablation of Ac also affects sphingomyelin levels

Our data show that Ac deficiency alters erythropoiesis, resulting in decreased frequencies of reticulocytes in peripheral blood and bone marrow of Ac KO mice. Nonetheless, during *P. yoelii* infection of Ac KO mice reticulocytes start to repopulate, which is also reflected in rapidly rising parasitemia 10 and 14 days p.i. (*Figure 2A*). To confirm that the induced Ac KO is persistent at later time points, we analyzed Ac expression in the bone marrow of noninfected mice on days 0 and 7 and 10 days after *P. yoelii* infection. As validated by RT-qPCR, Ac expression was still absent in Ac KO mice 7 and 10 days p.i. (*Figure 5A*).

In order to investigate whether other ceramidases compensate for Ac activity, we analyzed their mRNA expression. We found the neutral ceramidase (Nc) to be upregulated after induction of the Ac KO (*Figure 5B*). However, analysis of ceramide levels in bone marrow of Ac WT and Ac KO mice revealed significantly higher ceramide concentrations in bone marrow of Ac KO mice compared to Ac WT littermates 7 days p.i. (*Figure 5C*, right panel), similar to those measured on day 0 (*Figure 5C*, left panel). This suggests that elevated Nc expression in Ac KO mice is not sufficient to completely compensate for Ac deficiency with regard to the ceramide content.

Next, we determined sphingomyelin, sphingosine, and S1P levels in bone marrow of Ac WT and Ac KO mice. Sphingomyelin is generated from ceramide by the activity of sphingomyelin synthases and has been shown to increase the number of reticulocytes in mice (*Scaro et al., 1982*). While sphingomyelin levels were decreased in bone marrow of Ac KO mice compared to Ac WT mice on day 0 (*Figure 5D*, left panel), they were significantly enhanced on day 7 in the presence (*Figure 5D*, right panel) and absence (*Figure 5—figure supplement 1*) of the infection, correlating with increasing reticulocyte frequencies observed before. Moreover, Ac KO mice showed slightly increased sphingosine and significantly elevated S1P contents on day 0 in comparison to Ac WT littermates, while levels were similar 7 days p.i. (*Figure 5E and F*). This might be a consequence of increased expression of neutral ceramidase (*Figure 5B*). Taken together, these results indicate an altered SL metabolism in bone marrow of Ac KO mice that does not only affect ceramide levels but also leads to significantly elevated sphingomyelin concentrations 7 days p.i., which might account for the repopulation of reticulocytes.

## Carmofur treatment leads to alleviated parasitemia and reduced reticulocyte frequencies during *P. yoelii* infection

Since the genetic ablation of Ac strongly affected erythropoiesis and, as a consequence, the course of *P. yoelii* infection, we investigated whether pharmacological inhibition of Ac has the same effect. Studies by *Realini et al., 2013* showed that the antineoplastic drug carmofur is a potent inhibitor



**Figure 5.** Alterations of sphingolipid metabolism in bone marrow of acid ceramidase (Ac) knockout (KO) mice. (**A**) Ac KO validation in bone marrow of noninfected (d0) and *P. yoelii*-infected (d7 and d10) Ac KO mice and Ac wildtype (WT) littermates as control by RT-qPCR (n = 5–9). (**B**) Fold change of neutral ceramidase (*Asah2*) expression in bone marrow on day 0 (n = 6–7). (**C**) Ceramide, (**D**) sphingomyelin, (**E**) sphingosine, and (**F**) S1P levels in bone marrow of noninfected Ac WT and Ac KO mice on day 0 (left panels), and 7 days post infection (p.i.) (right panels) were determined by high-performance liquid chromatography mass spectrometry (HPLC-MS/MS) (n = 3–6). Data are presented as mean ± SEM. Statistical analyses were performed using Mann–Whitney $U$-test for (**A**), unpaired Student's $t$-test for (**B**) and (**F**), and two-way ANOVA, followed by Sidak's post-test for (**C**) and (**D**) (*p<0.05, **p<0.01, ***p<0.001, ****p<0.0001).

The online version of this article includes the following source data and figure supplement(s) for figure 5:

**Source data 1.** Alterations of sphingolipid metabolism in bone marrow of acid ceramidase (Ac) knockout (KO)

*Figure 5 continued on next page*

*Figure 5 continued*
mice.

**Figure supplement 1.** Sphingolipid level of noninfected acid ceramidase (Ac) knockout (KO) and Ac wildtype (WT) mice.

**Figure supplement 1—source data 1.** Sphingolipid level of noninfected acid ceramidase (Ac) knockout (KO) and Ac wildtype (WT) mice.

of Ac activity in mice. Therefore, C57BL/6 mice were treated with carmofur or vehicle and infected with *P. yoelii* (*Figure 6A*). Strikingly, carmofur-treated mice showed a strong reduction in parasitemia (*Figure 6B*). Moreover, spleen weights of carmofur-treated mice were significantly reduced on day 7 p.i. (*Figure 6C*).

In order to confirm Ac inhibition, we determined SL levels in bone marrow of carmofur- and vehicle-treated mice. Indeed, ceramide contents were significantly increased upon carmofur treatment in *P. yoelii*-infected mice 7 days p.i. (*Figure 6D*) and noninfected mice (*Figure 6—figure supplement 1A*) compared to control mice, whereas sphingosine, S1P, and sphingomyelin levels were similar in all groups.

To analyze whether the milder course of infection in carmofur-treated mice is also associated with altered reticulocyte frequencies, we analyzed blood of carmofur- and vehicle-treated mice on day 0 as well as 3, 7, and 10 days after *P. yoelii* infection. While there is a tendency of decreased reticulocytes on day 0, carmofur-treated mice showed significantly reduced reticulocyte frequencies compared to vehicle-treated mice on days 3, 7, and 10 p.i. (*Figure 6E*). Furthermore, carmofur treatment led to an altered erythropoiesis in bone marrow with lower relative numbers of erythroblasts and increased percentages of erythrocytes compared to vehicle control 3, 7, and 10 days p.i. (*Figure 6F*, *Figure 6—figure supplement 1D and E*). Similar results were obtained in noninfected carmofur- and vehicle-treated mice (*Figure 6B–E*), indicating that modulation in erythropoiesis occurs independent from the infection.

*Plasmodium* genomes do not encode candidate acid ceramidases (*Aurrecoechea et al., 2009*). In support of this notion, cell biological assays and metabolomic profiling revealed ceramide uptake from the infected erythrocyte and modest levels that remain unchanged upon asexual parasite replication (*Gulati et al., 2015*; *Haldar et al., 1991*). However, to exclude any potential direct effects of carmofur on intraerythrocytic development in vitro we performed a *P. falciparum* growth inhibition assay (*Figure 6G*). Asexual *P. falciparum* parasites (strain 3D7) were propagated under normal cell culture conditions, enriched for early ring stages and exposed for 48 hr to increasing concentrations of carmofur. As controls, 1% DMSO and the antimalarial drug chloroquine were included. As reported (*Vennerstrom et al., 1999*), the $IC_{50}$ concentration of chloroquine in the drug-sensitive 3D7 strain was determined at 22 nM (*Figure 6—figure supplement 2*). In marked contrast, the $IC_{50}$ value for carmofur was 39.75 µM, and even at 100 µM viable parasites were detected (*Figure 6G*, upper panel). For comparison, the $IC_{50}$ of primaquine, a liver stage-specific drug with no clinical use against blood stages, was reported as 2 µM (*Vennerstrom et al., 1999*). Moreover, the $IC_{50}$ of carmofur for Ac was determined at 29 nM (*Realini et al., 2013*). Hence, we conclude that the effects of carmofur can be solely attributed to the inhibition of host cell Ac.

## Carmofur as efficient therapy against *P. yoelii* infection

Since therapy in clinics is usually started after the infection has already manifested, we next analyzed the therapeutic potential of carmofur treatment. Therefore, we infected C57BL/6 mice with *P. yoelii* and started with carmofur treatment on day 5 p.i., when the infection was established (*Figure 7A*). As depicted in *Figure 7B*, mice showed significantly reduced reticulocyte frequencies after carmofur therapy 7, 10, and 14 days p.i. This was accompanied by a strongly alleviated parasitemia (*Figure 7C*). Moreover, spleen weights of carmofur-treated mice were decreased compared to vehicle control (*Figure 7D*). Although carmofur therapy seems to reduce the production of reticulocytes, carmofur-treated mice showed similar or even slightly higher RBC counts than vehicle-treated mice during infection, indicating that they did not suffer from a more severe anemia (*Figure 7E*).

In summary, our results provide evidence that deficiency or inhibition of Ac activity alters erythropoiesis, resulting in decreased reticulocyte frequencies associated with lower parasitemia during *P. yoelii* infection.



**Figure 6.** Carmofur treatment leads to decreased parasitemia with lower reticulocyte frequencies. (**A**) For pharmacological inhibition of acid ceramidase (Ac), C57BL/6 mice were either treated with 750 µg carmofur or with vehicle as control by daily intraperitoneal (i.p.) injection starting 1 day prior to *P. yoelii* infection. (**B**) Parasitemia of infected carmofur- and vehicle-treated mice was determined at indicated time points by microscopy of Giemsa-stained blood films (n = 10–11). (**C**) Spleen weight of *P. yoelii*-infected carmofur- and vehicle-treated mice 7 days post infection (p.i.) (n = 9). (**D**) Ceramide, sphingosine, S1P, and sphingomyelin levels in bone marrow of infected carmofur- and vehicle-treated mice 7 days p.i. were determined by high-performance liquid chromatography mass spectrometry (HPLC-MS/MS) (n = 6–9). (**E**) Frequencies of Ter119+CD71+ reticulocytes in blood of *P. yoelii*-infected carmofur- and vehicle-treated mice were analyzed by flow cytometry on days 0, 3, 7, and 10 (n = 12–18). (**F**) Percentages of erythroblasts (Eb, CD71+FSChigh), reticulocytes (Ret, CD71+FSClow), and mature erythrocytes (Ery, CD71-FSClow) in bone marrow of *P. yoelii*-infected carmofur- and vehicle-treated mice on day 7 (n = 9). (**G**) *P. falciparum* growth inhibition assay to test potential effects of carmofur on intraerythrocytic development

*Figure 6 continued on next page*

*Figure 6 continued*

in vitro. Micrographs of infected erythrocytes that were either nontreated or exposed 48 hr to increasing doses of carmofur (top rows) or chloroquine (bottom row). Shown are representative images of standard growth assays (left panel). Bars: 2 μm. Dose–response assay (48 hr) of carmofur or DMSO as control against asexual *P. falciparum* growth (starting at ring stages, n = 3 in triplicates). 50% inhibitory concentration (IC$_{50}$) is indicated. Results from 2–3 independent experiments are presented as mean ± SEM. Statistical analyses were performed using two-way ANOVA, followed by Sidak's post-test for (**A**), unpaired Student's *t*-test for (**C**) and (**F**), and Mann–Whitney *U*-test for (**D**) and (**E**) (*p<0.05, **p<0.01, ****p<0.0001).

The online version of this article includes the following source data and figure supplement(s) for figure 6:

**Source data 1.** Carmofur treatment leads to decreased parasitemia with lower reticulocyte frequencies.

**Figure supplement 1.** Sphingolipid level and erythropoiesis of carmofur-treated noninfected and *P. yoelii*-infected mice.

**Figure supplement 1—source data 1.** Sphingolipid level and erythropoiesis of carmofur-treated noninfected and *P. yoelii*-infected mice.

**Figure supplement 2.** *P. falciparum* growth inhibition assay with chloroquine.

**Figure supplement 2—source data 1.** *P. falciparum* growth inhibition assay with chloroquine.

## Discussion

By hydrolyzing ceramide into sphingosine, Ac is a key regulator of the SL metabolism. We and others showed that ceramide signaling is involved in the regulation of cell differentiation, activation, apoptosis, and proliferation (*Hose et al., 2019*; *Claus et al., 2009*; *Okazaki et al., 1989*). However, the impact of cell-intrinsic Ac activity and ceramide on the course of *Plasmodium* infection remains unclear. Here, we made use of Ac-deficient mice with ubiquitously increased ceramide levels to study the impact of endogenous ceramide on *P. yoelii* infection.

An adequate immune response is necessary to fight malaria-causing *Plasmodium* infections. It is well known that T cells play an important role during the blood stage of malaria (*Perez-Mazliah and Langhorne, 2014*). Our data show decreased expression of molecules associated with T cell activation in Ac KO mice in the early infection phase (*Figure 2D*). However, this observation is rather due to a low parasite load than due to reduced T cell activation caused by Ac-deficiency per se. As T cell-specific Ac ablation in Ac CD4cre KO mice did not affect the course of *P. yoelii* infection (*Figure 3B*), endogenous ceramide levels do not seem to have a significant impact on T cell activation in this model.

In addition to its characteristics mentioned before, ceramide has also been proposed to modulate erythropoiesis. While long-chain ceramides (C22, C24:0, and C24:1) are described to induce erythroid differentiation in bone marrow cells of rats (*Clayton et al., 1974*), the treatment with C2 ceramide or bacterial ceramide-generating sphingomyelinase resulted in the inhibition of erythropoiesis in human bone marrow cells (*Dallalio et al., 1999*). However, these studies were conducted with externally applied ceramide and exclusively in vitro. The role of cell-intrinsic ceramide on RBC development in vivo largely remained elusive. Our data indicate that loss of Ac activity and consequently increased ceramide levels dramatically affect erythrocyte maturation in vivo. Bone marrow of Ac KO mice contained almost no erythroblasts and reticulocytes on day 0 (*Figure 4D*, left panel). Well in line, studies by *Orsini et al., 2019* showed reduced frequencies of orthochromatophilic erythroblasts and absence of reticulocytes in a culture of human hematopoietic stem progenitor cells from umbilical cord blood in response to C2 ceramide or bacterial sphingomyelinase treatment. They propose an inhibition of erythropoiesis independent of apoptosis, mediated by the induction of myelopoiesis via the TNFα/neutral sphingomyelinase/ceramide pathway. In contrast, Signoretto and colleagues observed increased annexin-V staining and therefore elevated eryptosis of human erythrocytes in response to ceramide accumulation induced by treatment with the ceramidase inhibitor ceranib-2 (*Signoretto et al., 2016*). Although we did not detect a shift from erythropoiesis toward myelopoiesis on day 0 (data not shown), RBCs from Ac KO mice exhibited significantly reduced annexin-V staining compared to Ac WT controls after 24 and 72 hr of in vitro incubation (data not shown). Thus, our data suggest that lower reticulocyte frequencies are not due to increased apoptosis of RBCs. Nevertheless, erythroblasts and reticulocytes seem to repopulate bone marrow and blood of *P. yoelii*-infected Ac KO mice from day 7 p.i. on (*Figure 4B and D*, right panel). Since ceramide levels continued to increase until day 7 p.i. (*Figure 5C*), there was at least no sufficient compensation by other ceramidases, although mRNA expression of Nc was upregulated on day 0 (*Figure 5B*). However, while sphingomyelin levels were significantly reduced in bone marrow of Ac KO mice compared to Ac WT littermates on day 0 (*Figure 5D*, left panel), they were highly elevated on day 7 p.i. (*Figure 5D*, right panel) when erythroblasts and reticulocytes reappear. Sphingomyelin is generated from ceramide

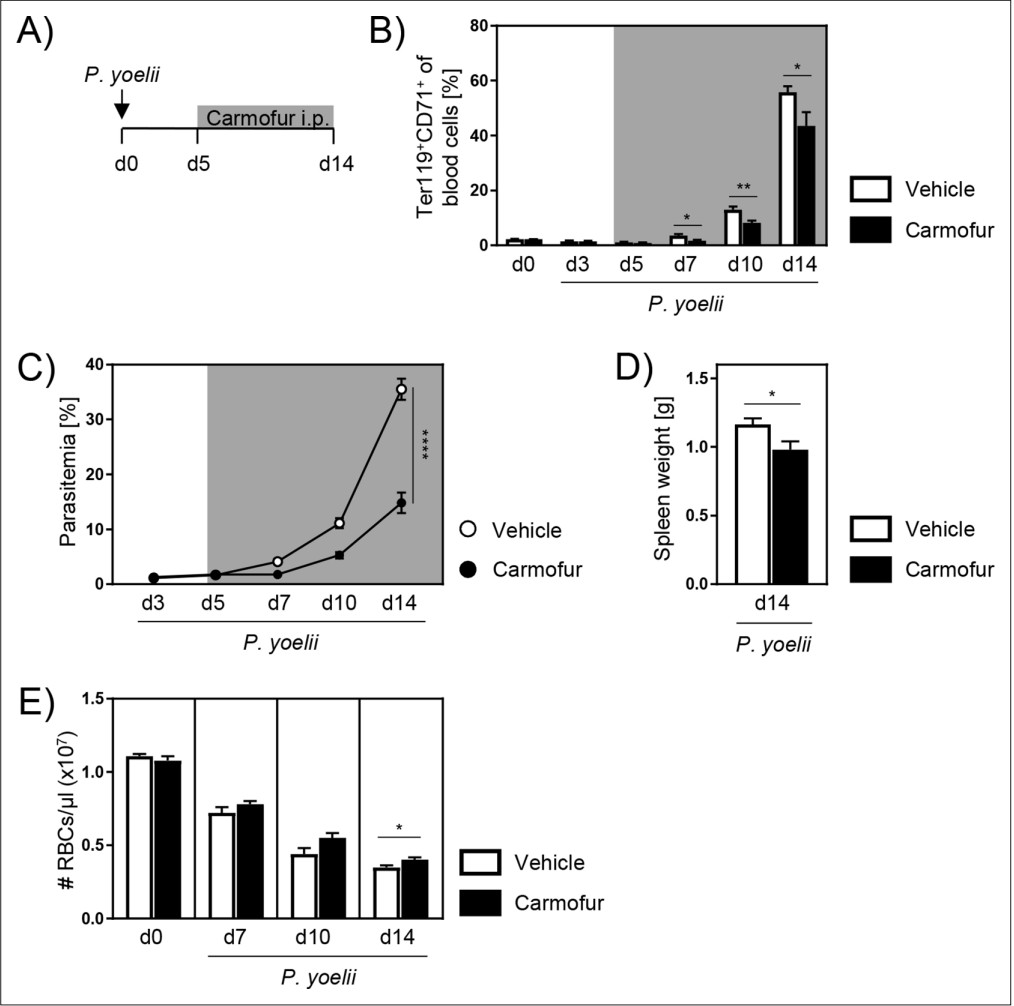

**Figure 7.** Carmofur application as therapeutic treatment of *P. yoelii* infection. (**A**) As therapeutic approach, C57BL/6 mice were infected with *P. yoelii* on day 0 and treated with 750 µg carmofur or vehicle only from day 5 post infection (p.i.) onward. (**B**) Percentages of Ter119⁺CD71⁺ reticulocytes in blood were analyzed by flow cytometry at indicated time points (n = 5–10). (**C**) Parasitemia of infected carmofur- and vehicle-treated mice was determined 3, 5, 7, 10, and 14 days p.i. by microscopy of Giemsa-stained blood films (n = 10). (**D**) Spleen weight of *P. yoelii*-infected carmofur- and vehicle-treated mice 14 days p.i. (n = 10). (**E**) Red blood cell (RBC) count per microliter blood of infected carmofur- and vehicle-treated mice was determined with an automated hematology analyzer (n = 5–10). Results from two independent experiments are summarized as mean ± SEM. Statistical analyses were performed using Mann–Whitney *U*-test for (**B**), two-way ANOVA, followed by Sidak's post-test for (**C**), and unpaired Student's *t*-test for (**D**) and (**E**) (*p<0.05, **p<0.01, ****p<0.0001).

The online version of this article includes the following source data for figure 7:

**Source data 1.** Carmofur application as therapeutic treatment of *P. yoelii* infection.

by sphingomyelin synthases, which are important regulators of intracellular and plasma membrane ceramide and sphingomyelin levels (*Li et al., 2007*). Consistent with our data, studies by *Clayton et al., 1974* showed an induction of erythropoiesis by sphingomyelin in bone marrow of rats in vitro. Moreover, Scaro and colleagues observed an increase of circulating reticulocytes after injecting mice with purified sphingomyelin in vivo (*Scaro et al., 1982*). Hence, one might speculate that the massive ceramide accumulation in Ac KO mice leads to an augmented SL turnover, resulting in elevated sphingomyelin concentrations in the bone marrow of Ac KO mice 7 days p.i. that might be responsible for resumption of erythropoiesis.

Several studies provide evidence that SLs and SL analogs are capable of regulating the development of *Plasmodium* parasites (*Labaied et al., 2004*; *Meyer et al., 2012*). C6 ceramide has been

shown to efficiently inhibit growth of *P. falciparum* parasites in a dose-dependent manner in vitro by reducing cytosolic glutathione levels and thereby inducing death of the parasites (*Pankova-Kholmyansky et al., 2003*). This cytotoxic effect was abolished by addition of S1P, a downstream metabolite of ceramide and sphingosine, respectively. The importance of S1P for parasite survival has recently also been analyzed by *Sah et al., 2021*. They demonstrated that *P. falciparum* parasites strictly depend on the erythrocyte endogenous S1P pool for their intracellular development. Inhibition of the S1P-generating enzyme sphingosine kinase-1 resulted in decreased glycolysis, which is essential for parasite energy supply. Interestingly, we measured significantly increased S1P contents in bone marrow of Ac KO mice on day 0 compared to Ac WT littermates, while levels were similar 7 days p.i. (*Figure 5F*). However, since Ac KO mice show highly reduced parasitemia, we would exclude that enhanced S1P levels beneficially affect parasite growth in our model. It is more likely that the lower parasite load in the early phase of infection is due to highly reduced reticulocyte frequencies in blood of Ac KO mice (*Figure 4B*), which represent the target cells of *P. yoelii* parasites (*Martín-Jaular et al., 2013*). Moreover, after reappearance of the reticulocytes, Ac KO an Ac WT mice exhibited similar parasitemia 10 and 14 days p.i. (*Figure 2A*).

As our data indicate the involvement of the Ac/ceramide axis in regulating erythropoiesis, which affects the course of *Plasmodium* infection, we speculated that these findings might open the door to new potential therapeutic strategies against malaria. The limited availability of antimalarial drugs and emerging parasite resistances demand the development of new approaches. Therefore, we made use of carmofur, an antineoplastic drug that has recently been described to inhibit Ac activity in vitro and in vivo (*Realini et al., 2013*; *Dementiev et al., 2019*). Carmofur-treated mice exhibited significantly reduced parasitemia and spleen weight compared to control mice during *P. yoelii* infection (*Figure 6B and C*). Consistent with our observations in Ac KO mice, carmofur treatment led to decreased reticulocyte frequencies and an altered RBC development (*Figure 6E and F*). Previous studies have shown the induction of reticulocytosis during the course of *P. yoelii* infection (*Martín-Jaular et al., 2013*; *Martin-Jaular et al., 2011*). As a direct effect of carmofur on the parasites might be possible, the question arises whether the lower parasite load in our model is due to the reduced reticulocyte count or, conversely, whether the lower parasitemia causes the lower reticulocytosis. Using a *P. falciparum* growth inhibition assay, we could demonstrate that the cytotoxic effect of carmofur is negligible (*Figure 6G*). Moreover, noninfected carmofur-treated mice show similar changes in erythropoiesis that we observed in the infected animals (*Figure 6—figure supplement 1B and C*). Hence, we assume that the effects of carmofur are solely attributed to the inhibition of murine Ac. However, possible side effects of carmofur should be considered when thinking about a clinical use as antimalarial therapy. Carmofur has been developed and widely used in Japan to treat patients suffering from carcinomas of the gastrointestinal tract and breast. Like other antineoplastic drugs, it has been reported to increase the risk of leukoencephalopathy, with an incidence of 0.026% between 1982 and 2008 (*Mizutani, 2008*). Nevertheless, also available antimalarial drugs have adverse effects, including cardiovascular toxicity, myopathy, and neurotoxicity, which should be considered (*AlKadi, 2007*).

In contrast to the murine model of blood-stage malaria, human *Plasmodium* infections often go along with low erythropoiesis and reticulocytosis, respectively (*Pathak and Ghosh, 2016*; *Chang et al., 2004a*). On the one hand, especially for reticulocyte-restricted species as *P. vivax*, this leads to a limited number of target cells for the parasites. On the other hand, it results in anemia that causes mortality and morbidity in patients. However, studies by Chang et al. and mathematical calculations by Cromer and colleagues revealed that these beneficial or detrimental effects depend on timing and the rate of reticulocyte production (*Chang et al., 2004b*; *Cromer et al., 2009*). Although carmofur-treated mice show a lower reticulocyte production and therefore lower parasitemia, they had similar or even higher RBC counts compared to vehicle-treated controls, meaning that they did not suffer from a more severe anemia (*Figure 7E*). Thus, carmofur treatment seems to beneficially influence the balance between parasite burden and RBC production. Finally, therapeutic carmofur administration to *P. yoelii*-infected mice also led to a decreased parasitemia and spleen weight (*Figure 7*), demonstrating the efficacy of carmofur therapy against malaria, even after manifestation of infection. Nevertheless, it should be considered that carmofur treatment might not be as effective against infections caused by *P. falciparum* with mixed tropism. Thus, future experiments using rodent *Plasmodium chabaudi* parasites, which are also able to infect both, reticulocytes and mature erythrocytes (*Leong et al., 2021*), could clarify this issue.

Overall, our results indicate that cell-intrinsic Ac activity plays an important role during erythropoiesis and therefore during blood-stage infections. Hence, pharmacological inhibition might represent a novel therapeutic strategy to conquer malaria and other infections caused by reticulocyte-prone pathogens.

# Materials and methods

## Mice

Mice were bred and housed under specific pathogen-free conditions at the animal facility of the University Hospital Essen. All experiments were performed in accordance with the guidelines of the German Animal Protection Law and approved by the State Agency for Nature, Environment, and Consumer Protection (LANUV), North Rhine-Westphalia, Germany (Az 84-02.04.2015.A474, Az 81-02.04.2018.A302). Female C57BL/6 mice were purchased from Envigo Laboratories (Envigo CRS GmbH). *Asah1/Rosa26^creER* mice (Asah1tm1Jhkh/Gt(Rosa)26Sortm9(cre/ESR1)Arte) express a mutated form of the estrogen receptor (ER) that is fused to the Cre recombinase (Taconic Biosciences Inc). Two loxp-sites are flanking exon 1 of the *Asah1* gene. Upon tamoxifen administration, the Cre recombinase is translocated into nucleus, leading to the deletion of exon 1 and therefore loss of function. *Asah1/Cd4^cre* mice (Asah1tm1Jhkh/Tg(CD4-cre)1Cwi) were generated by breeding *Asah1* mice with *Cd4^cre* mice (Tg(CD4-cre)1Cwi). The Cre recombinase is expressed under the control of CD4 regulatory elements. *Asah1/Lyz2^cre* mice (Asah1tm1Jhkh/Lyz2tm1(cre)Ifo) were obtained by crossing *Asah1* mice with *Lyz2^cre* mice (Lyz2tm1(cre)Ifo) that express Cre recombinase under the control of the lysozyme M promoter (Lyz2).

## Tamoxifen administration

Tamoxifen (Sigma-Aldrich) was dissolved in corn oil and administered to *Asah1/Rosa26^creER/+* mice (Ac KO) and *Asah1/Rosa26^+/+* littermates (Ac WT) by intraperitoneal (i.p.) injection (4 mg in 100 µl) 8, 6, and 4 days prior to analysis or infection.

## Carmofur treatment

C57BL/6 mice were either treated with 750 µg carmofur (Abcam) dissolved in 100 µl corn oil or with corn oil only (vehicle) by daily i.p. injection starting 1 day prior to infection. For therapeutic use of carmofur, C57BL/6 mice were treated with carmofur or vehicle 5 days p.i.

## *P. yoelii* infection

Cryopreserved *P. yoelii* 17XNL-infected red blood cells (iRBCs) were passaged once through C57BL/6 mice, before using them for infection of experimental mice. Experimental mice were infected with $1 \times 10^5$ *P. yoelii*-parasitized RBCs by intravenous (i.v.) injection on day 0. The parasitemia of infected mice was examined by microscopy of Giemsa-stained blood films. RBC counts were analyzed using the automated hematology analyzer KX-21N (Sysmex).

## Cell isolation and flow cytometry

Splenocytes were isolated by rinsing spleens with erythrocyte lysis buffer and washing with PBS supplemented with 2% FCS and 2 mM EDTA. Single-cell suspensions of bone marrow cells were generated by flushing bones (tibia) with PBS. Blood was either taken by puncture of the tail vein during the experiment or by cardiac puncture on day of sacrifice, and immediately mixed with heparin to avoid coagulation. Magnetic-activated cell sorting (MACS) was used to separate and enrich different cell populations from spleen, peritoneal lavage, or blood. Cells of interest were isolated using the CD4+/CD8a+ T cell isolation kit, the Neutrophil isolation kit, CD11c MicroBeads UltraPure, and F4/80 MicroBeads UltraPure (all Miltenyi Biotec) according to the manufacturer's instructions and separated automatically with the AutoMACS from Miltenyi Biotec. CD4+ and CD8+ T cells were additionally sorted by fluorescence-activated cell sorting (FACS) using an ARIA II cell sorter (BD Bioscience). For flow cytometry analyses, cells were stained with anti-CD4, anti-CD8, anti-CD49d (BD Bioscience), anti-Foxp3, anti-Ki67, anti-CD71 (eBioscience), anti-PD1, anti-CD11a, anti-CD45, and anti-Ter119 (BioLegend) conjugated to fluorescein isothiocyanate (FITC), pacific blue (PB), phycoerythrin (PE), allophycocyanin (APC), Alexa Flour 647 (AF647), or PE-cyanin 7 (PE-Cy7). The fixable viability dye

eFluor780 (eBioscience) was used to identify dead cells. Intracellular staining for Foxp3 and Ki67 was performed using the Fixation/Permeabilization kit (eBioscience) according to the manufacturer's protocol. Cells were analyzed by flow cytometry with an LSR II instrument (BD Bioscience) using DIVA software.

## Sphingolipid quantification by HPLC-MS/MS

Spleen tissue homogenates or cell suspensions were subjected to lipid extraction using 1.5 mL methanol/chloroform (2:1, v:v) as described before (*Gulbins et al., 2018*). The extraction solvent contained $d_7$-sphingosine ($d_7$-Sph), $d_7$-sphingosine 1-phosphate ($d_7$-S1P), C17 ceramide (C17 Cer), and $d_{31}$-C16 sphingomyelin ($d_{31}$-C16 SM) (all Avanti Polar Lipids) as internal standards. Chromatographic separations were achieved on a 1290 Infinity II HPLC (Agilent Technologies) equipped with a Poroshell 120 EC-C8 column (3.0×150 mm, 2.7 μm; Agilent Technologies). MS/MS analyses were carried out using a 6495 triple-quadrupole mass spectrometer (Agilent Technologies) operating in the positive electrospray ionization mode (ESI+) (*Naser et al., 2020*). The following mass transitions were recorded (qualifier product ions in parentheses): *long-chain bases:* $m/z$ 300.3 → 282.3 (252.3) for Sph, $m/z$ 307.3 → 289.3 (259.3) for $d_7$-Sph, $m/z$ 380.3 → 264.3 (82.1) for S1P, and $m/z$ 387.3 → 271.3 (82.1) for $d_7$-S1P; *ceramides:* $m/z$ 520.5 → 264.3 (282.3) for C16 Cer, $m/z$ 534.5 → 264.3 (282.3) for C17 Cer, $m/z$ 548.5 → 264.3 (282.3) for C18 Cer, $m/z$ 576.6 → 264.3 (282.3) for C20 Cer, $m/z$ 604.6 → 264.3 (282.3) for C22 Cer, $m/z$ 630.6 → 264.3 (282.3) for C24:1 Cer, and $m/z$ 632.6 → 264.3 (282.3) for C24 Cer; *sphingomyelins:* $m/z$ 703.6 → 184.1 (86.1) for C16 SM, $m/z$ 731.6 → 184.1 (86.1) for C18 SM, $m/z$ 734.6→ 184.1 (86.1) for $d_{31}$-C16 SM, $m/z$ 759.6 → 184.1 (86.1) for C20 SM, $m/z$ 787.7 → 184.1 (86.1) for C22 SM, $m/z$ 813.7 → 184.1 (86.1) for C24:1 SM, and $m/z$ 815.7 → 184.1 (86.1) for C24 SM. Peak areas of Cer and SM subspecies, as determined with MassHunter software (Agilent Technologies), were normalized to those of the internal standards (C17 Cer or $d_{31}$-C16 SM) followed by external calibration in the range of 1 fmol to 50 pmol on column. Sph and S1P were directly quantified via their deuterated internal standards $d_7$-Sph (0.25 pmol on column) and $d_7$-S1P (0.125 pmol on column). Determined SL amounts were normalized either to cell count or to the actual protein content (as determined by Bradford assay) of the tissue homogenate used for extraction.

## RNA isolation, cDNA synthesis, and qRT-PCR

RNA was either isolated using the RNeasy Fibrous Tissue Kit (QIAGEN) for spleen, thymus, and liver tissue, or using the RNeasy Mini Kit (QIAGEN) for bone marrow cells, T cells, macrophages, dendritic cells, and neutrophils according to the manufacturer's instructions. To synthesize cDNA, 0.1–1 μg of RNA was reversed transcribed using M-MLV Reverse Transcriptase (Promega) with dNTPs, Oligo-dT mixed with Random Hexamer primers (Thermo Fisher Scientific). For quantitative real-time PCR, Fast SYBR Green Master Mix (Thermo Fisher Scientific) was used on a 7500 Fast Real-Time PCR System (Thermo Fisher Scientific). Each sample was measured as technical duplicate. The expression levels of target genes were normalized against *ribosomal protein S9* (*RPS9*). The following primer sequences (5'–3') were used: *Asah1* (TTC TCA CCT GGG TCC TAG CC, TAT GGT GTG CCA CGG AAC TG), *Asah2* (AGA GAG AGC AAG GTA TTC TTC, ACT ATT CAC AAA GTG GTT GC), *RPS9* (CTG GAC GAG GGC AAG ATG AAG C, TGA CGT TGG CGG ATG AGC ACA).

## *P. falciparum* growth inhibition assay

Growth inhibition of cultured *P. falciparum* was done as published (*Smilkstein et al., 2004*) with modifications by *Spry et al., 2013*. Cultures of Pf3D7 master cells were used in flat-bottom 96-well plates. Carmofur (Abcam) was diluted to final concentrations of 0.3, 1, 3, 10, 30, and 100 μM. Chloroquine-diphosphate (Sigma-Aldrich) was diluted to 0.3, 1, 3, 10, 30, and 100 nM final concentrations as positive control. As negative control, the solvent DMSO (1 %) was used. The cultures were adjusted to a parasitemia of 1%, containing predominantly ring stage parasites, assessed by microscopic examination of Giemsa-stained films. The hematocrit was adjusted to the final value of 1% per 200 μl well. Cultures were incubated at 37°C for 48 hr with a gas mixture of 5% $O_2$ and 5% $CO_2$ in $N_2$. After 48 hr, samples were taken for Giemsa-staining and the plates transferred to –80°C for cell lysis. For the fluorescent assay, the plates were thawed and cells were resuspended with 1:1 SYBR Safe DNA Gel Stain (Thermo Fisher, final concentration: 1:10,000) and incubated in the dark at 37°C for 30 min. Fluorescence was measured with a Synergy HTX reader at 485 nm excitation wavelength and 428 nm

emission wavelength. To independently confirm the results of the fluorescence assay, parasites were monitored by standard light microscopy of the Giemsa-stained films at 1000× magnification. Arrangement and scaling of the images was done with the Affinity Photo 1.10.5.1342 program.

### Statistical analyses

Statistical analyses were performed using GraphPad Prism 7 software. To test for Gaussian distribution, D'Agostino–Pearson omnibus and Shapiro–Wilk normality tests were used. If data passed normality testing unpaired Student's $t$-test was performed, otherwise Mann–Whitney $U$-test was used. Differences between two or more groups with different factors were calculated using two-way ANOVA followed by Sidak's post-test. p-Values of 0.05 or less were considered indicative of statistical significance (*$p<0.05$, **$p<0.01$, ***$p<0.001$, ****$p<0.0001$).

## Acknowledgements

We kindly thank Sina Luppus and Christina Liebig for excellent technical assistance. We further thank Daniel Herrmann for the help with HPLC-MS/MS sphingolipid quantification. This work was supported by the Deutsche Forschungsgemeinschaft (DFG-GRK2098 to AMW, JB, KSL, EG, WH, and DFG-GRK2581 to BK).

## Additional information

### Funding

| Funder | Grant reference number | Author |
|---|---|---|
| Deutsche Forschungsgemeinschaft | GRK2098 | Karl Sebastian Lang |
| Deutsche Forschungsgemeinschaft | GRK2581 | Burkhard Kleuser |

The funders had no role in study design, data collection and interpretation, or the decision to submit the work for publication.

### Author contributions

Anne Günther, Conceptualization, Investigation, Visualization, Methodology, Writing - original draft; Matthias Hose, Conceptualization, Investigation, Methodology, Writing - review and editing; Hanna Abberger, Ylva Veith, Investigation; Fabian Schumacher, Investigation, Methodology, Writing - review and editing; Burkhard Kleuser, Karl Sebastian Lang, Erich Gulbins, Resources, Funding acquisition; Kai Matuschewski, Resources, Writing - review and editing; Jan Buer, Funding acquisition, data discussion; Astrid M Westendorf, Funding acquisition, data discussion; Wiebke Hansen, Conceptualization, Supervision, Funding acquisition, Project administration, Writing - review and editing

### Author ORCIDs

Matthias Hose http://orcid.org/0000-0003-0746-5591
Fabian Schumacher http://orcid.org/0000-0001-8703-3275
Kai Matuschewski http://orcid.org/0000-0001-6147-8591
Jan Buer http://orcid.org/0000-0002-7602-1698
Astrid M Westendorf http://orcid.org/0000-0002-2121-2892
Wiebke Hansen http://orcid.org/0000-0002-6020-0886

### Ethics

All experiments were performed in strict accordance with the guidelines of the German Animal Protection Law and approved by the State Agency for Nature, Environment, and Consumer Protection (LANUV), North Rhine-Westphalia, Germany (Az 84-02.04.2015.A474, Az 81-02.04.2018.A302).

### Decision letter and Author response

Decision letter https://doi.org/10.7554/eLife.77975.sa1
Author response https://doi.org/10.7554/eLife.77975.sa2

## Additional files

### Supplementary files
• Transparent reporting form

### Data availability
All data generated or analysed during this study are included in the manuscript and supporting file. Source Data files have been provided for all Figures and Figure Supplements.

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
