## [Editor Report]

This study provides evidence that murine acid ceramidase (Ac) is required for normal erythropoiesis and the development of rodent malaria. The findings are of interest in understanding molecular processes involved in regulating erythropoiesis, as well as the potential to develop host-directed therapies for malarial parasites that target human reticulocytes.

---

## [Decision Letter]

**Decision letter after peer review:**

Thank you for submitting your article "The acid ceramidase/ceramide axis controls parasitemia in *Plasmodium yoelii*-infected mice by regulating erythropoiesis" for consideration by *eLife*. Your article has been reviewed by 2 peer reviewers, including Malcolm J McConville as Reviewing Editor and Reviewer #1, and the evaluation has been overseen by Jos van der Meer as the Senior Editor.

Essential revisions:

1) The effect of carmofur treatment on parasite growth in vitro should be addressed.

2) The authors note that host sphingosine/S1P levels have been shown to modulate intraerythrocytic development of Plasmodium. Analysis of sphingosine/S1P levels in AcKO mice and/or carmofur treated mice is strongly recommended to verify whether changes in these lipids could also be involved in regulating parasite growth or retriculocyte levels.

*Reviewer #1 (Recommendations for the authors):*

Knock-out of Ac would also be expected to lead to changes in sphingosine and S1P levels, which have been shown to directly affect intraerythrocytic parasite development. Direct measurement of sphingosine/S1P levels in WT and Ac KO mice tissues using LC-MS would confirm whether the levels of these lipid do indeed change and potentially impact on the conclusions of the study.

The decrease in parasitemia and spleen weight seen in carmofur-treated animals is relatively modest. While the authors show that carmofur-treatment results in reduced reticulocyte numbers, it would be important to confirm the extent to which in vivo treatment inhibits Ac, by measuring levels of Cer, SM and sphingosine/S1P in drug-treated animals.

The effect of carmofur on intraerythrocytic development in vitro should be tested.

The authors suggest that Ac inhibitors could be used to control infections by reticulocyte targeting species of malaria. Could the authors comment on whether there is a risk that these inhibitors could exacerbate infections caused by *P. falciparum* given that carmofur-treatment leads to a modest increase in mature RBC levels?

The authors could reference other studies that have identified metabolic processes that are required for erythropoiesis and development of rodent malaria.

*Reviewer #2 (Recommendations for the authors):*

Parasitemia in Ac KO mice is lower at days 3 and 7 pi. Why do they catch up at later time points? Decreased T cell activation might reflect the decreased parasitemia. How is the T cell response at day 14 (when spleens are already as big as in WT mice)?

P. yoelii NL is self limiting. Why did the authors choose to only till day 14?

Naive Ac KO mice do have strongly reduced reticulocyte counts but were similar to WT mice upon infection on day 10. Can the authors comment on why the effect in naive mice is that strong but vanishes upon infection?

In Figure 6 the experiment is performed till day 14 but the analyses appears to be performed at selected time points. It would be great to see also other data on time points parallel to parasitemia (days 3, 7, 10, 14).

The effect of carmofur is striking but when discussing a potential therapeutic use side effects should be at least mentioned. The authors discussed a direct role on parasites but did they try to study a direct action on parasites?

in Figure 3 B the data of WT mice are missing (white balls).

In the legends different statistical tests were mentioned but often it remained unclear which test was used for which set of data.

---

## [Author Response]

Reviewer #1 (Recommendations for the authors):Knock-out of Ac would also be expected to lead to changes in sphingosine and S1P levels, which have been shown to directly affect intraerythrocytic parasite development. Direct measurement of sphingosine/S1P levels in WT and Ac KO mice tissues using LC-MS would confirm whether the levels of these lipid do indeed change and potentially impact on the conclusions of the study.

We thank the reviewer for this helpful suggestion. We now measured sphingosine and S1P levels in bone marrow cells of Ac WT and Ac KO mice at day 0 and 7 days p.i. and included the data in Figure 5 (E + F) of the revised manuscript. Interestingly, Ac KO mice showed slightly increased sphingosine and significantly elevated S1P levels at day 0, which might be a consequence of increased expression of neutral ceramidase (Figure 5 B). However, since S1P has been described to support parasite development and survival, but Ac KO mice have a reduced parasite load, we would exclude that these changes led to the differences in parasitemia observed between Ac WT and Ac KO mice.

The decrease in parasitemia and spleen weight seen in carmofur-treated animals is relatively modest. While the authors show that carmofur-treatment results in reduced reticulocyte numbers, it would be important to confirm the extent to which in vivo treatment inhibits Ac, by measuring levels of Cer, SM and sphingosine/S1P in drug-treated animals.

We agree that analysis of sphingolipid levels is necessary to confirm Ac inhibition by carmofur. Indeed, carmofur treatment led to significant increased ceramide content in bone marrow cells, while levels of sphingosine and S1P were not altered. Data for *P. yoelii*-infected (Figure 6 D) and non-infected (Figure 6 – figure supplement 1) carmofur- and vehicle-treated mice are now included in the revised manuscript.

The effect of carmofur on intraerythrocytic development in vitro should be tested.

The reviewer raised an important point. Genomic data suggest that Plasmodium parasites do not encode an acid ceramidase. However, to exclude any direct inhibitory effect on intra-erythrocytic parasite growth a drug response assay is warranted. As suggested, this is best done with cultured *P. falciparum* parasites. We have now performed this experiment, which is shown in Figure 6 G of the revised manuscript. For this assay, we have done two complementary assays, microscopic examination of cultured parasites and SYBR green staining measured by a fluorescence plate reader. The assays were done in three biological replicates with technical triplicates each. We determined an IC_50_ of 39.8 µM for carmofur, fully supporting the notion that the observed effects can be solely attributed to the inhibitory effect of carmofur on the murine acid ceramidase.

The authors suggest that Ac inhibitors could be used to control infections by reticulocyte targeting species of malaria. Could the authors comment on whether there is a risk that these inhibitors could exacerbate infections caused by *P. falciparum* given that carmofur-treatment leads to a modest increase in mature RBC levels?

We thank the reviewer for this helpful comment. We carefully discussed this point in the revised manuscript.

The authors could reference other studies that have identified metabolic processes that are required for erythropoiesis and development of rodent malaria.

We have now included information about metabolic processes in the introduction section of the revised manuscript.

Reviewer #2 (Recommendations for the authors):Parasitemia in Ac KO mice is lower at days 3 and 7 pi. Why do they catch up at later time points? Decreased T cell activation might reflect the decreased parasitemia. How is the T cell response at day 14 (when spleens are already as big as in WT mice)?

We thank the reviewer for these helpful comments. Our study shows that Ac-deficient mice have significantly reduced frequencies of reticulocytes in blood and bone marrow at early time points after KO induction. Since reticulocytes represent the target cells of *P. yoelii* parasites, Ac KO mice showed highly reduced parasitemia 3 and 7 p.i.. However, reticulocytes reappear at later time points (10 and 14 days p.i.) and are at a similar level to that of Ac WT mice, which is also reflected in the parasitemia. We hypothesize that reticulocytes return due to an augmented ceramide turnover to sphingomyelin 7 days p.i., which has been described to induce erythropoiesis (Clayton RB et al., J Lipid Res. 1974;15(6):557-62).

We totally agree that the reduced T cell activation is most likely due to less parasitemia and correspondingly less antigens. This is supported by similar T cell responses of Ac WT and Ac KO mice 14 days p.i., when both groups showed similar parasitemia. We now included these data as Figure 2 – figure supplement 1 in the revised manuscript.

P. yoelii NL is self limiting. Why did the authors choose to only till day 14?

The lack of Ac activity leads to the development of Farber disease, a severe lysosomal storage disorder. Homozygous mice with a mutant form of Ac show symptoms such as weight loss, slowed growth and enlargement of lymphoid organs starting at 5 weeks of age (Alayoubi AM et al., EMBO Mol Med 2013; 5: 827-42). Therefore, we performed experiments only until day 14 (3 weeks from first tamoxifen administration) to avoid a Farber disease phenotype that might affect our results.

Naive Ac KO mice do have strongly reduced reticulocyte counts but were similar to WT mice upon infection on day 10. Can the authors comment on why the effect in naive mice is that strong but vanishes upon infection?

The observed effect is independent of the infection. Upon KO induction by tamoxifen administration, reticulocytes are significantly reduced in Ac KO mice compared to Ac WT littermates. In both, *P. yoelii*-infected (Figure 4 D) and non-infected (Figure 4 – figure supplement 1) Ac KO mice reticulocytes reappear at day 7. As described above, we speculate that increasing sphingomyelin levels in bone marrow of Ac KO mice at later time points are responsible for the resumption of erythropoiesis. We have also included data on sphingomyelin levels for non-infected mice in Figure 5 – figure supplement 1.

In Figure 6 the experiment is performed till day 14 but the analyses appears to be performed at selected time points. It would be great to see also other data on time points parallel to parasitemia (days 3, 7, 10, 14).

We thank the reviewer for this helpful comment. We now included additional data for day 3 and 10 in parallel to experiments performed with Ac KO mice (Figure 4) in Figure 6 and Figure 6 – figure supplement 1 of the revised manuscript.

The effect of carmofur is striking but when discussing a potential therapeutic use side effects should be at least mentioned. The authors discussed a direct role on parasites but did they try to study a direct action on parasites?

We thank the reviewer for this helpful suggestions. We have now carefully discussed possible side effects. As already mentioned in response to reviewer #1 (comment 3), we have now analysed the direct effect of carmofur on *Plasmodium* parasites. Data are included in Figure 6 G of the revised manuscript.

In Figure 3 B the data of WT mice are missing (white balls).

We apologize and thank the reviewer for this comment. Indeed, the parasitemia of Ac CD4cre WT and Ac CD4cre KO mice were quite similar and therefore the white balls were hidden by the black ones. We now additionally added bars to the graph to avoid this confusion.

In the legends different statistical tests were mentioned but often it remained unclear which test was used for which set of data.

We apologize for this mistake. As described in the material and methods part, we tested data for Gaussian distribution by D’Agostino-Pearson omnibus and Shapiro-Wilk normality tests. If data passed normality testing unpaired Student’s t-test was performed, otherwise Mann-Whitney U-test was used. Differences between two or more groups with different factors were calculated using two-way ANOVA followed by Sidak’s post-test. We have now included detailed information about statistical tests in the figure legends.